# The Optimization of the Interior Permanent Magnetic Motor Case Study

**Bogdan Mociran * and Vasile Topa**

Faculty of Electrical Engineering, Technical University of Cluj-Napoca, 28 Memorandumului Street, 400114 Cluj-Napoca, Romania
* Correspondence: bogdan.mociran@ethm.utcluj.ro

**Abstract:** This paper presents two optimization methods of an interior permanent magnetic motor (IPM). The first method is based on the non-dominated sorting genetic algorithm II (NSGA-II), while the second one is based on the NSGA-III algorithm. The focus is on the reduction of the cogging torque (CT), which acts upon the rotor when the IPM is not powered, and the current that flows through the three windings is 0. The influence of the CT over the motor operation is a negative one, inducing vibrations that are not desired. The limitation of these influences is the desired way for improvement. The optimizations applied entailed the change of the geometrical configuration of the stator whilst maintaining the exterior dimensions unaltered. Through this simple but robust approach, the performance of the IPM proposed was improved.

**Keywords:** interior permanent magnetic motor; cogging torque; optimization

## 1. Introduction

The increasing frequency of use of the IPM in all areas of activity, from industrial ones such as electric machines to household appliances like washing machines or the air conditioning units, requires a more silent and less vibrating way of functioning to support visual and hearing comfort for the users, Ref. [1] proposed an optimal shape design of the iron core to reduce CT in IPM motors. Their study investigated the effects of core geometry on the CT and provided valuable insights into improving motor performance. The way IPM works makes it a smart choice for a multitude of machinery because the use of permanent magnets makes it more robust from a framework point of view and because it does not require increased accuracy when the internal components are being built. Ref. [2] focused on torque ripple optimization by introducing skewing techniques for field weakening operation. Their research aimed to enhance the dynamic performance of IPM motors in various operating conditions.

The increase in efficiency of an IPM approach can be looked at from various angles—for instance, one that entails the optimization of the rotor, another being the optimization of the stator, or even a general optimization. Rotor optimization has also been extensively explored. Ref. [3] considered mechanical stress in their design optimization process to ensure the structural integrity of the rotor. Ref. [4] investigated rotor shape optimization based on magnetization direction, aiming to improve motor efficiency and performance. Ref. [5] proposed novel double-barrier rotor designs to reduce torque pulsation, thereby enhancing the overall motor performance. Furthermore, advancements have been made in winding configurations. Ref. [6] examined the application of concentrated windings in IPM machines and its impact on motor performance. In a subsequent study, Ref. [7] conducted parameter analysis of an IPM machine with fractional-slot-concentrated windings, focusing on open-circuit analysis to assess its electrical characteristics. The study detailed in [8] proposes a stator optimization focusing on the introduction of a number of dummy slots in the stator geometry, this number being adjusted depending on what is aimed to be improved. The general optimization of the IPM gained increased relevance in the initial

device design process, and with the technology advancements today, through the boosting of computational power and the use of fast computing methods, IPM complex geometries can be perfected. Ref. [9] proposed a hybrid method for fast computation of airgap flux and magnetic forces, enabling efficient design optimization of IPM motors. Moreover, computational tools such as finite element analysis (FEA) have been utilized for modeling and design optimization of IPM machines. Refs. [10,11] developed a computationally efficient FEA approach in optimization. They demonstrated its efficacy in achieving accurate and reliable results. Making use of the genetic algorithms' great capacity to handle numerous objective functions that have multiple constraints, it led to a considerable improvement in the internal configuration of the IPM. Ref. [12] adopted a two-stage design approach for multi-objective optimization of IPM motors used in compressor applications. Their research demonstrated the effectiveness of this approach in achieving enhanced motor performance for specific applications. Ref. [13] proposed a novel flux-concentrating rotor design combining a Halbach Permanent Magnet (Halbach PM) array and a spoke-type IPM machine. Their study focused on multi-objective optimization to improve motor efficiency and torque density. Ref. [14] introduced an asymmetric interior permanent magnet synchronous machine (AIPMSM), which exhibited improved torque characteristics compared to traditional symmetric designs. Ref. [15] investigated the use of an asymmetrical V-shape rotor configuration to enhance torque characteristics in IPM machines. Their research showed promising results in achieving improved performance through rotor design optimization. An approach based on mathematical calculus makes the translation of the IPM's internal components into numerical equations was presented in [16], which focused on reduction in time spent. Part of the latest trend is a method that tries to resolve the many-objective optimization problems and details applied to the IPM optimization by the authors of [17]. The current paper takes into account the setting-up and calibration of a test model based on the example from [18]. Ref. [19] focuses on the sequential subspace optimization design of a dual three-phase permanent magnet synchronous hub motor. It specifically highlights the utilization of the NSGA-III algorithm for the optimization process. The research provides detailed insights into the methodology, data analysis, and results, demonstrating the effectiveness of the proposed approach in enhancing the performance of electric motors. The authors' contribution in the interconnection process presented in our previous work [20] lies in the development and implementation of a novel framework that facilitates the seamless integration and communication between designing programs (such as computer-aided design (CAD) software), simulation tools (such as FEA software), and optimization methods (such as genetic algorithms). This framework, which incorporates several programs and algorithms, enables engineers to leverage the strengths of these tools, improving the efficiency and effectiveness of the design process.

In this paper, we aim to address the issue of CT in an IPM motor through two optimization methods: the NSGA-II and the NSGA-III algorithm incorporated in the system from [20]. Our contribution lies in applying these methods to improve IPM performance by reducing CT and consequently minimizing unwanted vibrations. By modifying the stator's geometrical configuration while keeping the external dimensions unchanged, we achieved significant results in terms of CT reduction and enhancing the IPM motor's operation.

## 2. Materials and Methods

The model from Figure 1a shows a halved view of the IPM, and it is characterized as follows: 4 poles and 12 stator slots with a constant length of 0.5 mm between the stator and the rotor. The blueprints can be found on the (b) side of the same figure, i.e., Figure 1b, where: in black, it displays the 8 dimensions in (mm) that are subjected to optimization where area of each winding was kept at 50 mm$^2$; in grey, the constant dimensions in (mm) and the total depth of 150 mm. The H–B curve describes the degree of magnetism of the ferromagnetic material utilized in the construction of the rotor and the stator. This curve is modelled from the data used in [18] in order to suit the requirements of the Comsol simulation program, the characteristic is displayed in Figure 2a. On the (b) side, the

characteristic attributed to the permanent magnets is displayed. The coercive field has the value of Hc = $7.957 \times 10^5$ Am$^{-1}$, while the permanent magnets are represented by B0 = 1 T, the horizontal ones being oriented towards the outer layer and the vertical ones being oriented towards the inner layer.

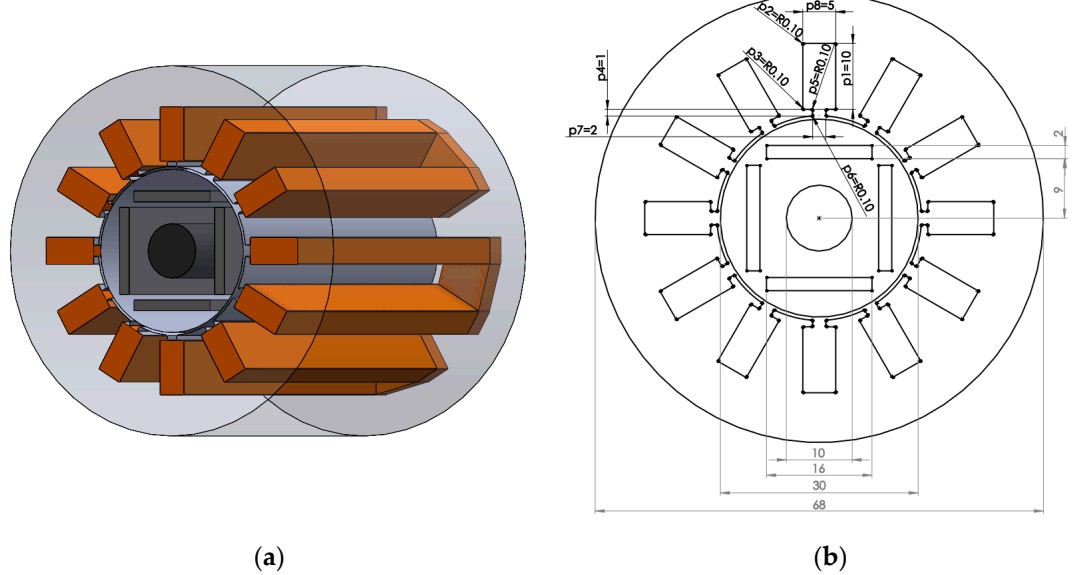

(**a**)  (**b**)

**Figure 1.** (**a**) A perpendicular halved section of a 3D model, the motor shaft in black, the permanent magnets in dark grey, the windings in orange, and the stator and rotor in light grey; (**b**) the 2D section with the initial values (in mm) of the simulation model, the optimization parameters in black, the constant values in grey.

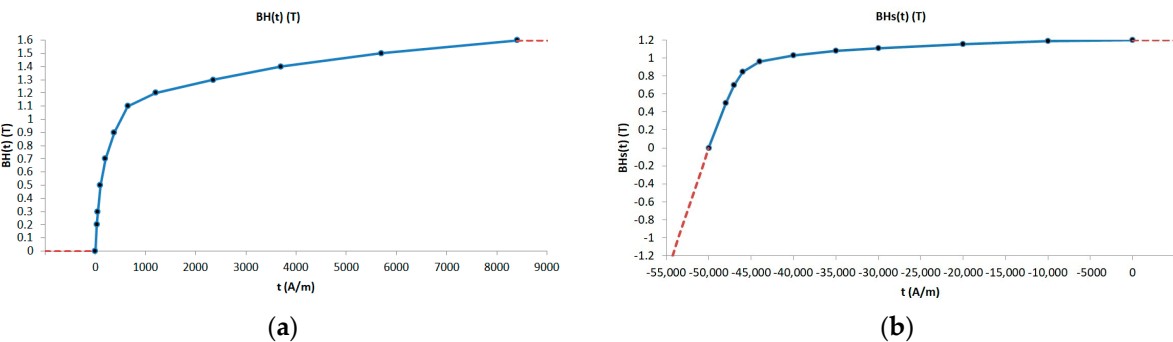

(**a**)  (**b**)

**Figure 2.** (**a**) The magnetization curve used for describing the behavior of the materials used for the stator and rotor; (**b**) the characteristic of the permanent magnets. Here the blue line follows the curve described by the input values, while the red marks represent the extrapolations of these values.

Taking into account the aforementioned values, the initial model was created, correlated, and calibrated with the data from [18] regarding the torque generated by the IPM in the no-load operation. In the first phase, in the model tackled for correlation, the optimization parameters were not included (the parameters that were exemplified in black in Figure 1), aiming at obtaining a more precise model. To be mentioned here is the fact that the model is a 3D one in which the influence of the ends of windings are taken into consideration, while these were not considered in the approach proposed by [18].

The representation of the magnetic induction map is displayed in Figure 3, where the two starting models are regarded. In the isometric view, only half of the IPM is shown due to the fact that the model symmetry was used to considerably reduce (by 50%) the time spent simulating the model.

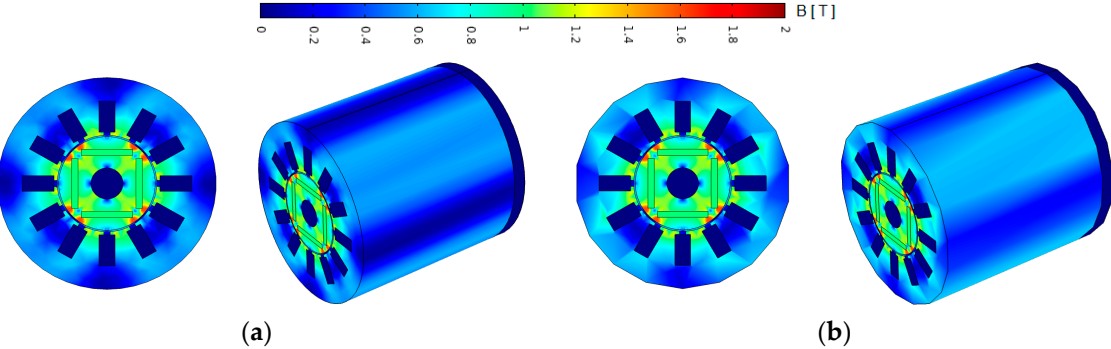

(**a**)         (**b**)

**Figure 3.** Magnetic induction map: (**a**) transversal and isometric views of the 1/2 of the initial model, without the optimization parameters; (**b**) transversal and isometric views of the 1/2 of the optimization model, with the angles rounded set for optimization.

In order to calculate the values of the torque, the tool used was the one available in Comsol for force calculation based on the Maxwell stress tensor. The data obtained by this tool regarding the torque values produced by the two IPMs compared to the existing data are presented in Figure 4. It can be observed that for a 90° functioning cycle, the values obtained are similarly scaled to those existing in [18], which demonstrates the accuracy of the calibration of the model—not only the initial model but also the setup for improvement. From the graph below, it can be noticed, as expected, that the functioning of the IPM in the no-load operation in a cyclical one, the period for completion being 30°. Taking this into consideration, in the following undertakings, the execution will be reduced to a single period where the average toque value is 0. Figure 5 presents torque values from 0–30°.

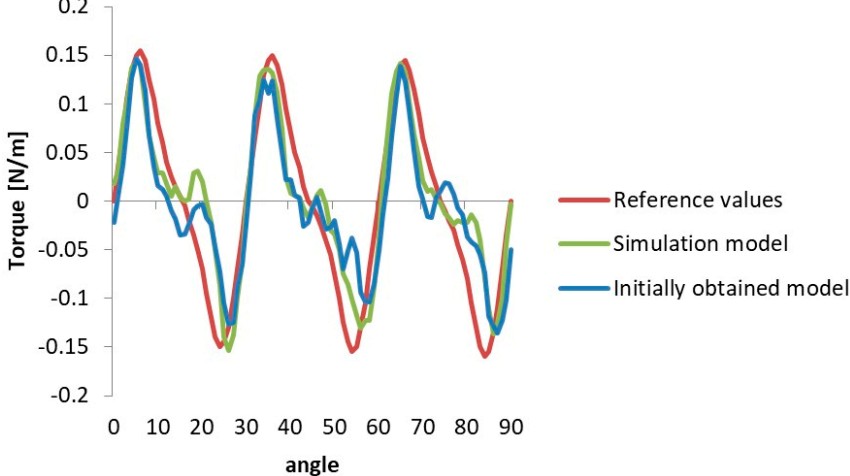

**Figure 4.** Torque values of the IPM for the execution from 0–90°. The red curve represents the results of the research from Ref. [18].

NSGA-II is an extension of the original NSGA algorithm. It employs a genetic algorithm framework combined with a non-dominated sorting and crowding distance mechanism to guide the search towards the Pareto front. The Pareto front represents the set of optimal solutions in which no solution can be improved in one objective without sacrificing performance in another.

The schematic representation of the NSGA-II algorithm is depicted in Figure 6. The interpretation is as follows: (1) Generate an initial population of candidate solutions. (2) Evaluate the objective function values for each candidate solution. (3) Assign a rank to each candidate solution based on their dominance relationships. (4) Calculate the crowding distance metric to maintain diversity within each rank. (5) Select individuals based on their rank and crowding distance to create the mating pool. (6) Apply genetic operators

such as crossover and mutation to create offspring solutions. (7) Create the next generation by combining the parent and offspring solutions using elitism. (8) Repeat Steps 2–7 until a termination condition is met (e.g., maximum number of generations or convergence criteria) [21].

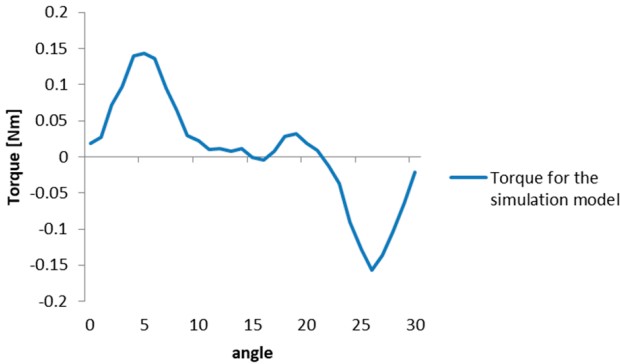

**Figure 5.** Torque values of the IPM in the simulation model for the execution from 0–30°.

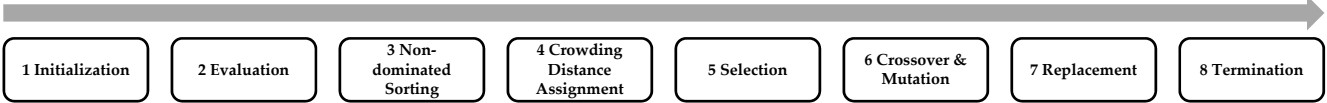

**Figure 6.** The schematic representation of NSGA-II.

NSGA-III is an enhancement of NSGA-II that addresses the limitation of NSGA-II in handling many-objective optimization problems. It introduces an improved selection scheme and a reference point-based approach for maintaining the diversity of solutions across the Pareto front. In Figure 7, the schematic representation of the NSGA-III algorithm is shown. The algorithm is executed as follows: (1) Generate an initial population of candidate solutions. (2) Evaluate the objective function values for each candidate solution. (3) Define a set of reference points uniformly distributed on the objective space. (4) Assign a rank to each candidate solution based on their dominance relationships. (5) Assign each candidate solution to the nearest reference point. (6) Select a diverse set of solutions from each reference point based on dominance and crowding distance. (7) Select individuals from the environmental selection for reproduction to create the mating pool. (8) Apply genetic operators such as crossover and mutation to create offspring solutions. (9) Update the reference points based on the solutions found in the current generation. 10 Repeat Steps 2–9 until a termination condition is met [22].

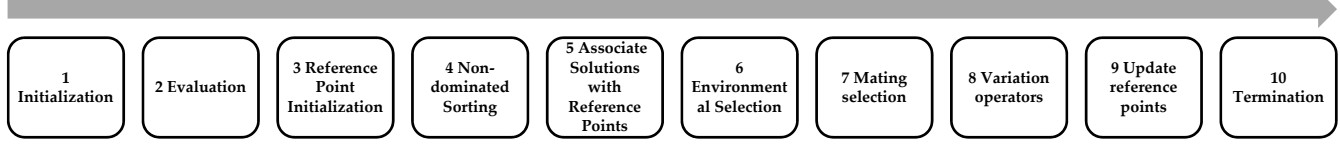

**Figure 7.** The schematic representation of NSGA-III.

The objective is to diminish the peak-to-peak values during the execution of the no-load operation, values that fall between 0.15 and −0.15 [Nm]. The objective functions resulted from the analysis of the graph, and they entail the attempt to decrease the extreme values through the seizing of the maximal and minimal values that would be used as feed to the optimization algorithms as quantities of the objective functions.

Taking into account the definition of the optimized parameters in (mm) and considering the numbering provided in Figure 1b, the objective functions has been defined as follows:

The objective function $f_1(p_1, p_2, \ldots, p_8)$ is defined to minimize the maximum value of cogging torque within the interval $[0, \pi/12]$, while the objective function $f_2(p_1, p_2, \ldots, p_8)$ is defined to maximize the minimum value of cogging torque within the interval $[\pi/12, \pi/6]$.

$$
\begin{cases}
f_1(p_1, p_2, \ldots p_8) = \max_{\text{extreme}}\{y(p_1, p_2, \ldots p_8)|\ \theta\ \in\ [0,\ \pi/12]\} \\
f_2(p_1, p_2, \ldots p_8) = -\min_{\text{extreme}}\{y(p_1, p_2, \ldots p_8)|\ \theta\ \in\ [\pi/12,\ \pi/6)\} \\
\quad\quad 5.5\ \leq\ p_1\ \leq 16 \\
\quad\quad 0.2\ \leq\ p_2\ \leq 1.5 \\
\quad\quad 0.2\ \leq\ p_3\ \leq 1.5 \\
\quad\quad\quad 1\ \leq\ p_4\ \leq 2 \\
\quad\quad 0.2\ \leq\ p_5\ \leq 0.5 \\
\quad\quad 0.2\ \leq\ p_6\ \leq 0.5 \\
\quad\quad 0.5\ \leq\ p_7\ \leq 7 \\
\quad\quad\quad 3\ \leq\ p_8\ \leq 9
\end{cases}
\tag{1}
$$

where $y(p_1, p_2, \ldots, p_8)$ represents the cogging torque function with respect to the geometric parameters $p_1, p_2, \ldots, p_8$; $\max_{\text{extreme}}(y(p_1, p_2, \ldots, p_8))$ is the maximum value of cogging torque in the interval $[0, \pi/12]$; and $\min_{\text{extreme}}(y(p_1, p_2, \ldots, p_8))$ is the minimum value of cogging torque in the interval $[\pi/12, \pi/6]$.

A brief description of the method employed in this paper and presented in [20] encompasses the following steps: designing the 3D model in SolidWorks (Dessault Systems, Paris, France) importing the model into COMSOL Multiphysics for electromagnetic simulations, analyzing the obtained results and evaluating the device's performance, optimizing the model using sorting algorithms and genetic algorithms, and modifying the geometry and structure of the device based on the obtained results. This method streamlines the process of designing and simulating 3D devices, enabling the rapid achievement of optimal solutions. By employing this working method, the optimization process was undertaken.

The starting settings for both algorithms were identical ones. The number generated was set to 5, and population was set to 50 individuals. As per the graph in Figure 5, the function presents positive values above the horizontal axis, which leads to the intent of minimizing these values, the objective being the reduction towards 0. For the function that deals with negative values, when values fall below the horizontal axis, the objective is to maximize these values with the intention of bringing them closer to 0.

## 3. Results

### 3.1. The First Optimization

The time needed for the optimization process from the first iteration to the resulted optimized solution was 10 h and 11 min. The algorithm ran for 286 generations before it managed to converge towards the optimal value. In Figure 8, it can be deduced that the objective functions were stabilized after approximately 100 generations run by the algorithm. The reduction of the objective functions values is considerable; the values drop around 0.05 Nm.

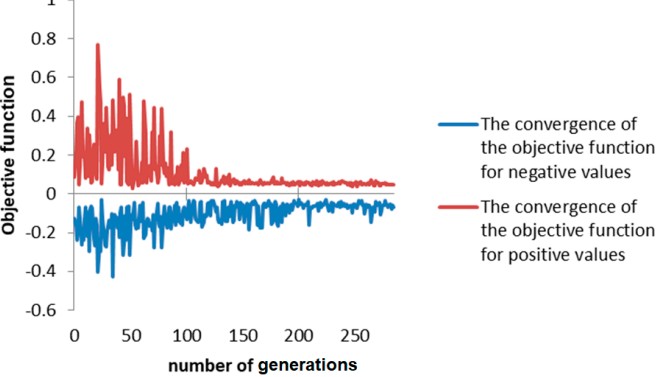

**Figure 8.** The values of the objective functions.

Figure 9 visually displays the values of the optimization parameters improved by the algorithm. The initial configuration suffers alterations through a chamfering of the edges, as expected, as well as elongation of the stator slots in comparison with the initial dimensions. On the (b) side of Figure 9, a transversal section of 1/2 of the optimized model is shown, over which a magnetic induction map was overlaid in the first step, i.e., 0°.

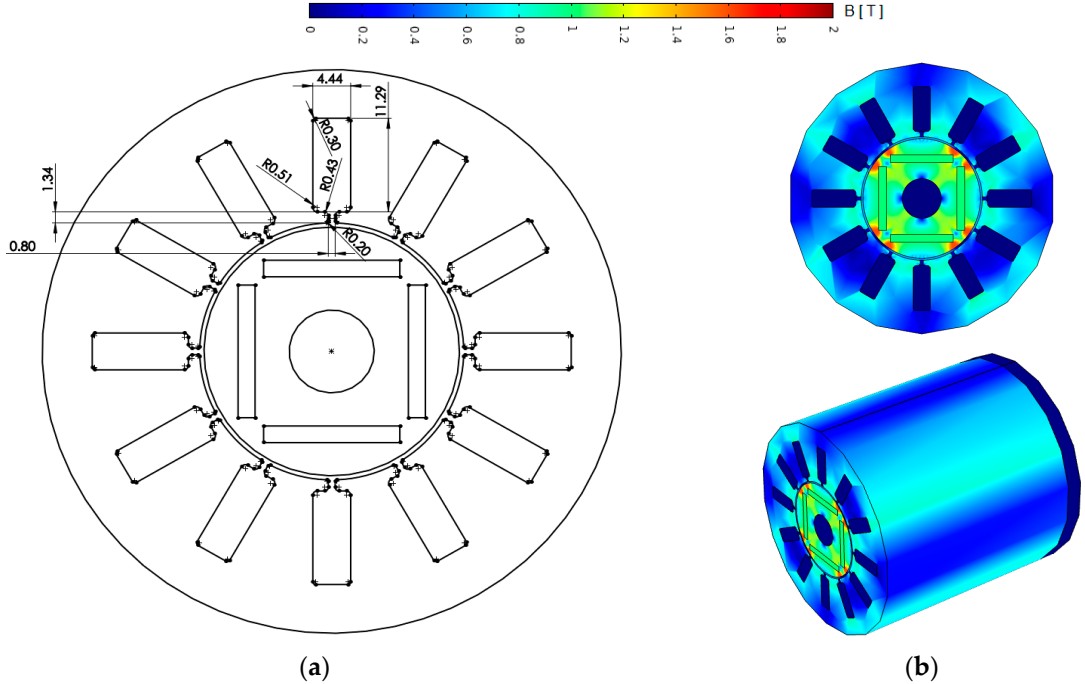

(**a**)    (**b**)

**Figure 9.** (**a**) Two-dimensional section showing the values (in mm) obtained after applying NSGA-II; (**b**) the magnetic-induction-map-optimized configuration.

The final result for the torque values, concerning the aforementioned configuration and regarding the execution for a period between 0–30° of the IPM, are highlighted in Figure 10. The conclusion is, upon rendering the results of the graph, that there is a distortion from the initially obtained sinusoidal shape can be observed as well as a significant decrease in peak values, which leads to a better functioning when used in on-load operation.

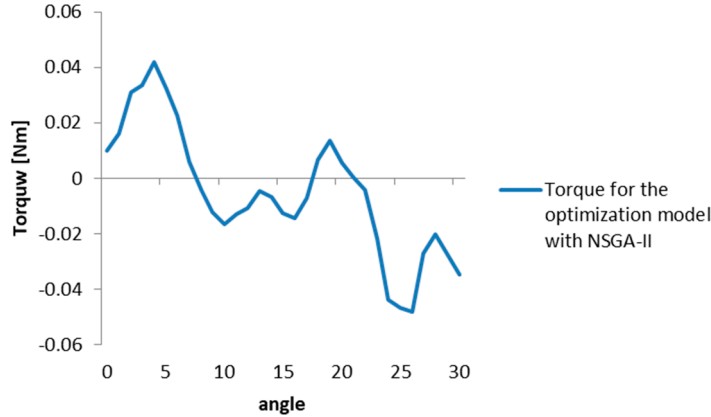

**Figure 10.** The torque values of the IPM optimized with NSGA-II.

### 3.2. The Second Optimization

The second optimization of the initial simulation model entailed the application of the NSGA-III algorithm as an optimization system, which ran for 306 generations over 10 h and 56 min. Moreover, the convergence occurred after approximately 100 generations,

which can be seen in Figure 11. In this figure, the two functions converge towards the optimal configurations around the value of 0.05 Nm, as in the previous case.

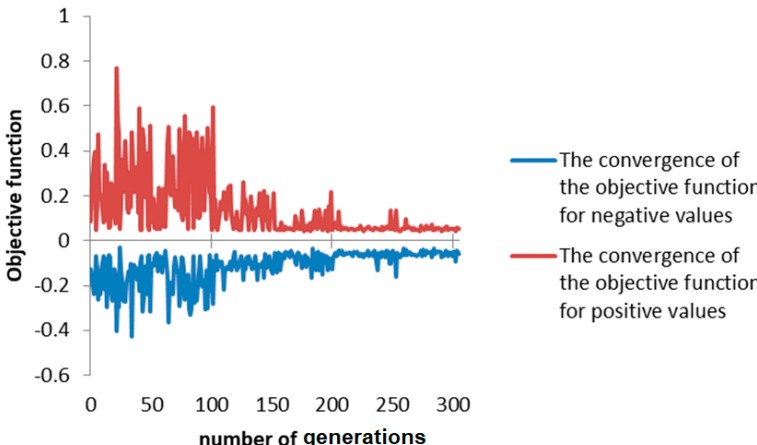

**Figure 11.** The values of the objectives functions.

For the consistency of the data, in Figure 12, the optimal configuration is presented in a 2D section with the mentioning of the optimized parameters. On the (b) side, the magnetic induction map in isometric view and cross-section for the improved IPM are shown.

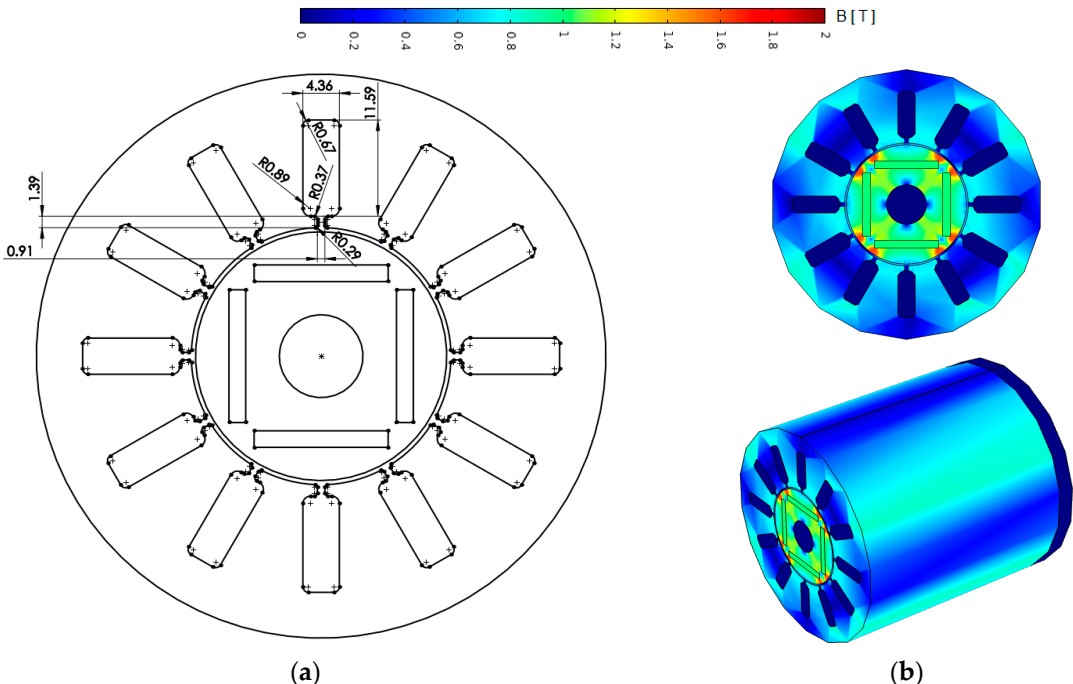

        (**a**)                  (**b**)

**Figure 12.** (**a**) Two-dimensional section showing the values (in mm) obtained after applying NSGA-III; (**b**) the magnetic induction map optimized configuration.

The final result for the torque values after applying the NSGA-III algorithm, as seen in Figure 13, shows a major decrease vis-à-vis the initial starting values as well as a distortion from the initially obtained sinusoidal shape. The change in shape can be seen as a consequence of the altered parameters, with the benefit of the decrease in the peak-to-peak torque values.

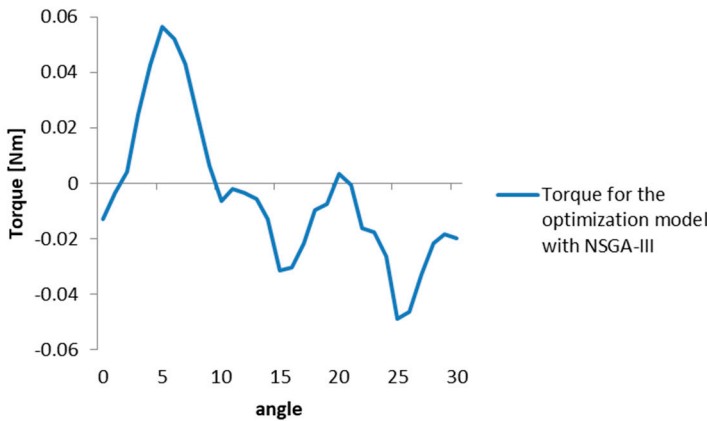

**Figure 13.** The torque values of the IPM optimized with NSGA-III.

## 4. Discussion

Starting from the same initial configuration, after applying both optimization systems, a significant flattening of the CT values was achieved by modifying eight internal dimensions of the stator. Figure 14 shows the magnitude by which the two new configurations (red for NSGA-II and green for NSGA-III) reduce this characteristic compared to the initial output (blue).

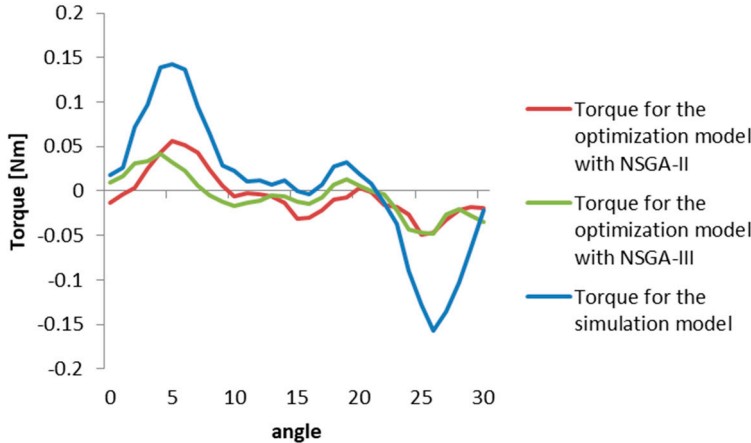

**Figure 14.** The torque values of the IPM for the simulation model configuration, for the model optimized with NSGA-II, and for the model optimized with NSGA-III during operation in the 0–30° ranges.

For a better understanding of the geometric differences between the configurations, Figure 15 on the left-hand side shows an overlap of the three models, where it can be observed that the rotor geometry remains constant, as well as the external dimensions of the IPM. A striking similarity between the two configurations obtained through optimization is highlighted. The right-hand side shows a detailed view of a stator slot, in which the improved geometries are also highlighted (in red for NSGA-II and green for NSGA-III), as well as the starting model in blue. The two configurations differ only by a small elongation of the stator slot and a tighter curvature at the top for the model presented in red. The differences in measured values between the initial model and the two optimized models can also be observed, highlighted in gray in the same picture. The three parameters with the highest influence for both algorithms in terms of improving the objective functions are marked in black.

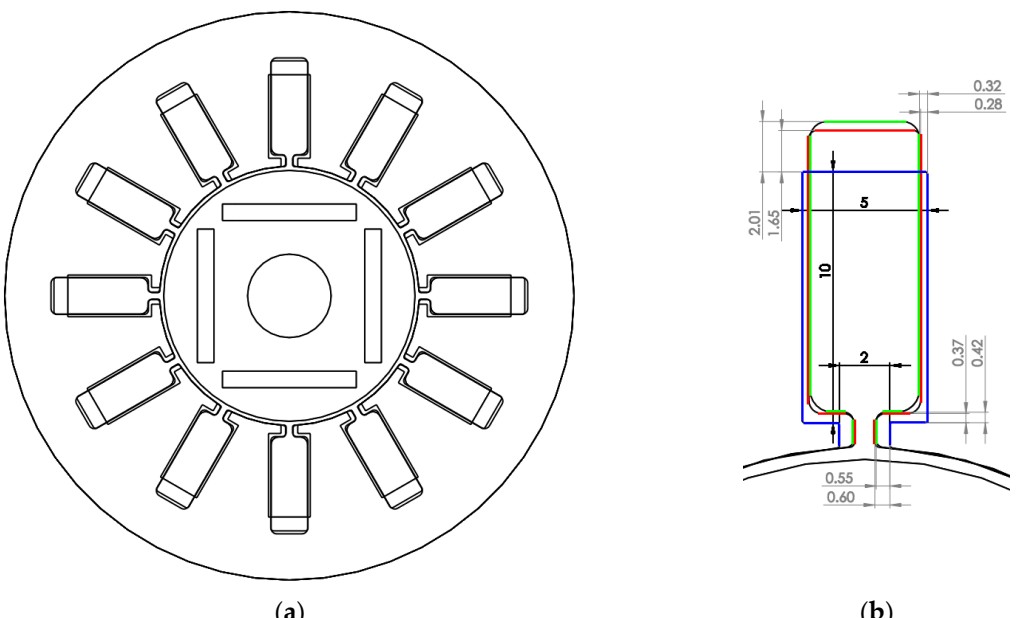

(**a**)　　　　　　　　　　　　　　　　(**b**)

**Figure 15.** (**a**) Two-dimensional section of the models; (**b**) detail compared to the simulation model configuration (blue), the model optimized with NSGA-II (red), and the model optimized with NSGA-III (green) (dimensions in mm).

## 5. Conclusions

Through the adjustment of specific parameters in the internal geometry of the IPM motor, a remarkable reduction in CT was achieved. The optimized models exhibited CT values of approximately 0.05 Nm, significantly lower than the initial model's value of 0.15 Nm. The findings suggest that the IPM motor design has a narrow range of potential constructive variations in terms of optimization. Despite this limitation, the vibrations and noise generated during motor operation can still be considerably reduced. The results indicate that the problem converges towards a specific geometric configuration, with minimal changes observed in the shape of the stator slot across different optimization outcomes. The study highlights the effectiveness of adjusting only eight parameters of the internal geometry to obtain robust configurations and consistent results for both approaches. In summary, the research demonstrates the potential for improving the performance of IPM motors by optimizing their internal geometry. The reduction in CT and the narrow range of constructive variations suggest that further enhancements can be made to minimize vibrations and noise, leading to more efficient and quieter motor operation.

Further research in the field of IPM motors should focus on exploring additional design parameters, conducting comparative analyses with alternative motor designs and noise reduction techniques, assessing the robustness of IPM motors under varying conditions, developing multi-objective optimization approaches, and investigating application-specific performance. By pursuing these directions, advancements can be made in the design, optimization, and application of IPM motors, leading to improved performance, reduced vibrations and noise, and increased efficiency.

**Author Contributions:** Conceptualization, B.M.; methodology, B.M.; validation, B.M. and V.T.; formal analysis, V.T.; investigation, B.M.; resources, V.T.; writing—original draft preparation, B.M.; writing—review and editing, V.T.; visualization, B.M.; supervision, V.T.; project administration, V.T.; funding acquisition, V.T. All authors have read and agreed to the published version of the manuscript.

**Funding:** This research received no external funding.

**Data Availability Statement:** Not applicable.

**Acknowledgments:** This paper was supported by the project "Entrepreneurial skills and research excellence in doctoral and postdoctoral study programs"—ANTREDOC (POCU/380/6/13/123927 CODE SMIS 123927).

**Conflicts of Interest:** The authors declare no conflict of interest.

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
