# Peer review of "The Optimization of the Interior Permanent Magnetic Motor Case Study"

_electronics, doi:10.3390/electronics12132982_

Round 1

Reviewer 1 Report

this work is well presented, the results are presented well discussed. However:

In introduction, the contribition is not clarified.

In introduction, the literature study is poor.

I propose adding a list of abbreviations and symbols.

The conclusion isn’t accuracy. 

Author Response

In introduction, the contribition is not clarified.

In this paper, we aim to address the issue of CT in an IPM motor through two optimization methods: the NSGA-II and the NSGA-III algorithm. Our contribution lies in applying these methods to improve IPM performance by reducing CT and consequently minimizing unwanted vibrations. By modifying the stator's geometrical configuration while keeping the external dimensions unchanged, we achieved significant results in terms of CT reduction and enhancing the IPM motor's operation

In introduction, the literature study is poor.

[1] proposed an optimal shape design of the iron core to reduce CT in IPM motors. Their study investigated the effects of core geometry on the CT and provided valuable insights into improving motor performance

[2] focused on torque ripple optimization by introducing skewing techniques for field weakening operation. Their research aimed to enhance the dynamic performance of IPM motors in various operating conditions

Rotor optimization has also been extensively explored. [3] considered mechanical stress in their design optimization process to ensure the structural integrity of the rotor. [4] investigated rotor shape optimization based on magnetization direction, aiming to improve motor efficiency and performance. [5] proposed novel double-barrier rotor designs to reduce torque pulsation, thereby enhancing the overall motor performance. Furthermore, advancements have been made in winding configurations. [6] examined the application of concentrated windings in IPM machines and its impact on motor performance. In a subsequent study, [7] conducted parameter analysis of an IPM machine with fractional-slot concentrated windings, focusing on open-circuit analysis to assess its electrical characteristics.

[9] proposed a hybrid method for fast computation of airgap flux and magnetic forces, enabling efficient design optimization of IPM motors. Moreover, computational tools such as finite element analysis have been utilized for modeling and design optimization of IPM machines. [10] developed a computationally efficient finite element analysis approach in optimization. They demonstrated its efficacy in achieving accurate and reliable results.

[12] adopted a two-stage design approach for multi-objective optimization of IPM motors used in compressor applications. Their research demonstrated the effectiveness of this approach in achieving enhanced motor performance for specific applications. [13] proposed a novel flux-concentrating rotor design combining a Halbach Permanent Magnet (Halbach PM) array and a spoke-type IPM machine. Their study focused on multi-objective optimization to improve motor efficiency and torque density. [14] introduced an asymmetric interior permanent magnet synchronous machine (AIPMSM), which exhibited improved torque characteristics compared to traditional symmetric designs. [15] investigated the use of an asymmetrical V-shape rotor configuration to enhance torque characteristics in IPM machines. Their research showed promising results in achieving improved performance through rotor design optimization.

I propose adding a list of abbreviations and symbols.

Thank you for the suggestion to include a list of abbreviations and symbols. However, I believe that by defining each abbreviation before using it within the text itself is sufficient for understanding, without the need for a separate list.

The conclusion isn’t accuracy.

Through the adjustment of specific parameters in the internal geometry of the IPM motor, a remarkable reduction in CT was achieved. The optimized models exhibited CT values of approximately 0.05 [Nm], significantly lower than the initial model's value of 0.15 [Nm]. The findings suggest that the IPM motor design has a narrow range of potential constructive variations when it comes to optimization. Despite this limitation, the vibrations and noise generated during motor operation can still be considerably reduced. The results indicate that the problem converges towards a specific geometric configuration, with minimal changes observed in the shape of the stator slot across different optimization outcomes. The study highlights the effectiveness of adjusting only eight parameters of the internal geometry to obtain robust configurations and consistent results for both approaches. In summary, the research demonstrates the potential for improving the performance of IPM motors by optimizing their internal geometry. The reduction in CT and the narrow range of constructive variations suggest that further enhancements can be made to minimize vibrations and noise, leading to more efficient and quieter motor operation.

Reviewer 2 Report

The mentioned research paper explores two optimization methods for reducing cogging torque in an interior permanent magnet motor (IPM). The methods, based on NSGA-II and NSGA-III algorithms, aim to improve the motor's performance by minimizing the undesirable vibrations caused by cogging torque.

The paper emphasizes the significance of mitigating cogging torque's negative impact on the motor's operation. It proposes optimizing the stator's geometrical configuration while maintaining the exterior dimensions unchanged as an effective approach. This modification enhances the IPM's performance, resulting in reduced cogging torque and improved operational characteristics.

In conclusion, the research paper focuses on utilizing optimization algorithms and design adjustments to minimize cogging torque and enhance the performance of an interior permanent magnet motor.

Comments:

Figure 10 is incorrectly labeled as Figure 2.

The paper has a strong citation base, although it appears that not all significant recent works have been included. An example of this is X. Sun, N. Xu, and M. Yao's publication titled "Sequential Subspace Optimization Design of a Dual Three-Phase Permanent Magnet Synchronous Hub Motor Based on NSGA III" in IEEE Transactions on Transportation Electrification, vol. 9, no. 1, pp. 622-630, March 2023, doi: 10.1109/TTE.2022.3190536, which focuses on three-phase motors. I include it mainly due to its highly detailed analysis of the obtained data and the presentation of results.

"In terms of presenting the results, it would be a good idea to enhance the format by standardizing the style of the graphs, which would improve readability. Similarly, the final effect could be quantitatively presented, such as indicating the geometric changes suggested by the algorithm, accompanied by commentary on the factors that may have influenced them.

The presentation of the research methods involves citing publications on the topic and sequentially presenting the successive steps. It may be worth briefly outlining the algorithm of operation to provide readers with a better understanding of this information.

The article presents a very concise technical form, which is its advantage due to the clear and precisely conveyed information. The authors utilized certain tools for conducting simulations. To make the entire presentation appear more scholarly, it would be worthwhile to emphasize the authors' contribution to the development of this technique of conducting simulation studies, which can be highlighted in the introduction or the conclusions.

Author Response

Figure 10 is incorrectly labeled as Figure 2.

It was a typo

The paper has a strong citation base, although it appears that not all significant recent works have been included. An example of this is X. Sun, N. Xu, and M. Yao's publication titled "Sequential Subspace Optimization Design of a Dual Three-Phase Permanent Magnet Synchronous Hub Motor Based on NSGA III" in IEEE Transactions on Transportation Electrification, vol. 9, no. 1, pp. 622-630, March 2023, doi: 10.1109/TTE.2022.3190536, which focuses on three-phase motors. I include it mainly due to its highly detailed analysis of the obtained data and the presentation of results.

[19] focuses on the sequential subspace optimization design of a dual three-phase permanent magnet synchronous hub motor. It specifically highlights the utilization of the NSGA III algorithm for the optimization process. The research provides detailed insights into the methodology, data analysis, and results, demonstrating the effectiveness of the proposed approach in enhancing the performance of electric motors.

"In terms of presenting the results, it would be a good idea to enhance the format by standardizing the style of the graphs, which would improve readability. Similarly, the final effect could be quantitatively presented, such as indicating the geometric changes suggested by the algorithm, accompanied by commentary on the factors that may have influenced them.

The differences in measured values between the initial model and the two optimized models can also be observed, highlighted in gray in the same picture. The three parameters with the highest influence for both algorithms in terms of improving the objective functions are marked in black. The update images are in the paper.

The presentation of the research methods involves citing publications on the topic and sequentially presenting the successive steps. It may be worth briefly outlining the algorithm of operation to provide readers with a better understanding of this information.

A brief description of the method employed in this paper and presented in [20] en-compasses the following steps: designing the 3D model in SolidWorks, importing the model into COMSOL Multiphysics for electromagnetic simulations, analysing the ob-tained results and evaluating the device's performance, optimizing the model using sorting algorithms and genetic algorithms, and modifying the geometry and structure of the device based on the obtained results. This method streamlines the process of designing and simulating 3D devices, enabling the rapid achievement of optimal solutions. By employing this working method, the optimization process was undertaken.

The article presents a very concise technical form, which is its advantage due to the clear and precisely conveyed information. The authors utilized certain tools for conducting simulations. To make the entire presentation appear more scholarly, it would be worthwhile to emphasize the authors' contribution to the development of this technique of conducting simulation studies, which can be highlighted in the introduction or the conclusions.

The authors' contribution in the interconnection process presented in our previous work [20] lies in the development and implementation of a novel framework that facilitates the seamless integration and communication between designing programs (such as Computer-Aided Design CAD software), simulation tools (such as FEA software), and optimization methods (such as genetic algorithms). This framework, which incorporates several programs and algorithms, enables engineers to leverage the strengths of these tools, improving the efficiency and effectiveness of the design process.

Reviewer 3 Report

This paper proposed optimization methods of an interior permanent magnetic motor (IPM) to reduce the cogging torque (CT), which acts upon the rotor when the IPM is not powered and the current that flows through the three windings is 0. The authors reported a good improvement. However, this reviewer has the following suggestions for the improvement of the article:

·         Enhancing the literature survey section by discussing the advantages and disadvantages of the recent and relevant articles is highly recommended.

·         In the materials and methods section, the author discussed the IPM but did not discuss the NSGA-II and NSGA-III methods. Please provide brief discussions on the optimization techniques, including their flow charts.

·         It is also highly recommended to add the formulation of the optimization problems.

·         Figures 6 and 9 are suspicious; if the algorithms are correctly coded, the objective values should not be greater in the later generations. To avoid such issues, the authors can use the ‘elitism’ operations of the genetic algorithm. Also, please avoid the term ‘steps’; write ‘generations’ instead.

·         Please provide a comparative analysis (existing literature vs. your results).

·         It also strongly recommended implementing the proposed strategy in other configurations of the IPM (changing the number of poles, stators, etc.).

·         Finally, please guide the readers about the future extension of this research in the conclusion section.

Overall, the article is interesting and can be considered after major revision, as suggested. 

 Moderate editing of English language required

Author Response

Enhancing the literature survey section by discussing the advantages and disadvantages of the recent and relevant articles is highly recommended.

The literature review section has been enhanced by incorporating the valuable suggestions provided by the reviewers. By addressing the reviewers' suggestions, the literature survey section now offers a more well-rounded and informative analysis of the chosen articles.

In the materials and methods section, the author discussed the IPM but did not discuss the NSGA-II and NSGA-III methods. Please provide brief discussions on the optimization techniques, including their flow charts.

NSGA-II is an extension of the original NSGA algorithm. It employs a genetic algorithm framework combined with a non-dominated sorting and crowding distance mechanism to guide the search towards the Pareto front. The Pareto front represents the set of optimal solutions where no solution can be improved in one objective without sacrificing performance in another.

The schematic representation of the NSGA-II algorithm is depicted in Figure 6. The interpretation is as follows: 1 Generate an initial population of candidate solutions. 2 Evaluate the objective function values for each candidate solution. 3 Assign a rank to each candidate solution based on their dominance relationships. 4 Calculate the crowding distance metric to maintain diversity within each rank. 5 Select individuals based on their rank and crowding distance to create the mating pool. 6 Apply genetic operators such as crossover and mutation to create offspring solutions. 7 Create the next generation by combining the parent and offspring solutions using elitism. 8 Repeat steps 2-7 until a termination condition is met (e.g., maximum number of generations or convergence criteria) [21].

Figure 6. The schematic representation of NSGA-II

NSGA-III is an enhancement of NSGA-II that addresses the limitation of NSGA-II in handling many-objective optimization problems. It introduces an improved selection scheme and a reference point-based approach for maintaining the diversity of solutions across the Pareto front. In Figure 7, the schematic representation of the NSGA-III algorithm is shown. The algorithm is executed as follows: 1 Generate an initial population of candidate solutions. 2 Evaluate the objective function values for each candidate solution. 3 Define a set of reference points uniformly distributed on the objective space. 4 Assign a rank to each candidate solution based on their dominance relationships. 5 Assign each candidate solution to the nearest reference point. 6 Select a diverse set of solutions from each reference point based on dominance and crowding distance. 7 Select individuals from the environmental selection for reproduction to create the mating pool. 8 Apply genetic operators such as crossover and mutation to create offspring solutions. 9 Update the reference points based on the solutions found in the current generation. 10 Repeat steps 2-9 until a termination condition is met [22].

Figure 7. The schematic representation of NSGA-III

It is also highly recommended to add the formulation of the optimization problems.

Taking into account the definition of the optimized parameters in (mm) and considering the numbering provided in Figure 1(b), the objective functions has been defined as follows:

The objective function f₁(p₁,p₂,… p₈) is defined to minimize the maximum value of cogging torque within the interval [0, π/12], while the objective function f₂(p₁,p₂,… p₈) is defined to maximize the minimum value of cogging torque within the interval  [π/12, π/6].

You can find the definition of objective functions  in the paper.

where y(p₁,p₂,… p₈) represents the cogging torque function with respect to the    geometric parameters p₁,p₂,… p₈,

maxextreme(y(p₁,p₂,… p₈)) is the maximum value of cogging torque in the interval [0, π/12],

minextreme(y(p₁,p₂,… p₈)) is the minimum value of cogging torque in the interval [π/12, π/6].

Figures 6 and 9 are suspicious; if the algorithms are correctly coded, the objective values should not be greater in the later generations. To avoid such issues, the authors can use the ‘elitism’ operations of the genetic algorithm. Also, please avoid the term ‘steps’; write ‘generations’ instead.

The concerns raised about the former Figures 6 and 9 are valid. The initial idea was to use the unaltered algorithms compared to the standard version to see if, under the given conditions, they would yield satisfactory results using the Black Box principle.

Please provide a comparative analysis (existing literature vs. your results).

Thank you kindly for your suggestion. I greatly appreciate your input. However, I would like to mention that the research presented here is just a part of the authors' endeavor to develop an optimization system based on the integration of multiple available programs and algorithms, with the aim of streamlining work in the field of electrical engineering. It is important to note that the research does not solely rely on a single model, specifically the one presented in this study. Instead, the main idea revolves around the application of optimizations to some initial models and observing the obtained results, rather than conducting a specific comparative study solely focused on this particular model.

It also strongly recommended implementing the proposed strategy in other configurations of the IPM (changing the number of poles, stators, etc.).

The suggestion of implementing the proposed strategy in other configurations of the IPM (changing the number of poles, stators, etc.) has been duly noted and is already being considered. While the current study focuses on a specific configuration, we recognize the importance of exploring the applicability and effectiveness of the strategy across different configurations. However, due to the scope and resources available for the current research, the focus is primarily on the chosen configuration. Nevertheless, the valuable suggestion will be incorporated into our future research endeavors, where we can further investigate and analyze the proposed strategy in various IPM configurations. This will allow for a comprehensive evaluation of its performance and potential benefits in a wider range of scenarios. Thank you for your insightful suggestion, and we will strive to expand our investigations in future studies.

Finally, please guide the readers about the future extension of this research in the conclusion section.

Further research in the field of IPM motors should focus on exploring additional design parameters, conducting comparative analyses with alternative motor designs and noise reduction techniques, assessing the robustness of IPM motors under varying conditions, developing multi-objective optimization approaches, and investigating application-specific performance. By pursuing these directions, advancements can be made in the design, optimization, and application of IPM motors, leading to improved performance, reduced vibrations and noise, and increased efficiency.

Round 2

Reviewer 3 Report

This reviewer would like to thank the respected authors for their efforts in updating the article. It can be accepted.

Minor English editing is required.